# Characterization of Human B Cell Hematological Malignancies Using Protein-Based Approaches

**DOI:** 10.3390/ijms25094644

**Published:** 2024-04-24

**Authors:** Cristina Jiménez, Alba Garrote-de-Barros, Carlos López-Portugués, María Hernández-Sánchez, Paula Díez

**Affiliations:** 1Hematology Department, University Hospital of Salamanca (HUS/IBSAL), CIBERONC and Cancer Research Institute of Salamanca-IBMCC (USAL-CSIC), 37007 Salamanca, Spain; jscris@usal.es; 2Department of Biochemistry and Molecular Biology, Pharmacy School, Universidad Complutense de Madrid, 28040 Madrid, Spain; albgarro@ucm.es (A.G.-d.-B.); marher36@ucm.es (M.H.-S.); 3Department of Translational Hematology, Instituto de Investigación Hospital 12 de Octubre (imas12), Hematological Malignancies Clinical Research Unit H12O-CNIO, 28029 Madrid, Spain; 4Department of Physical and Analytical Chemistry Chemistry, Faculty of Chemistry, University of Oviedo, 33006 Oviedo, Spain; uo269482@uniovi.es; 5Health Research Institute of the Principality of Asturias (ISPA), 33011 Oviedo, Spain; 6Department of Functional Biology, Faculty of Medicine and Health Science, University of Oviedo, 33006 Oviedo, Spain

**Keywords:** B cell disorder, lymphoma, leukemia, multiple myeloma, protein, flow cytometry, mass cytometry, mass spectrometry

## Abstract

The maturation of B cells is a complex, multi-step process. During B cell differentiation, errors can occur, leading to the emergence of aberrant versions of B cells that, finally, constitute a malignant tumor. These B cell malignancies are classified into three main groups: leukemias, myelomas, and lymphomas, the latter being the most heterogeneous type. Since their discovery, multiple biological studies have been performed to characterize these diseases, aiming to define their specific features and determine potential biomarkers for diagnosis, stratification, and prognosis. The rise of advanced -omics approaches has significantly contributed to this end. Notably, proteomics strategies appear as promising tools to comprehensively profile the final molecular effector of these cells. In this narrative review, we first introduce the main B cell malignancies together with the most relevant proteomics approaches. Then, we describe the core studies conducted in the field and their main findings and, finally, we evaluate the advantages and drawbacks of flow cytometry, mass cytometry, and mass spectrometry for the profiling of human B cell disorders.

## 1. Introduction to B Cell Hematological Disorders

B cell hematologic malignancies encompass a spectrum of cancers originating from abnormal B lymphocytes that affect the peripheral blood (PB), bone marrow (BM), and lymphatic system. They can be broadly categorized into leukemias, lymphomas, and myelomas (Figure 1). Each has distinct characteristics, depending on the type of cells affected, and different behaviors, ranging from chronic conditions with slow progression to aggressive forms that require immediate treatment. Understanding the alterations, key protein markers (Table 1), molecular pathways, and microenvironments that contribute to these diseases is crucial for their accurate diagnosis and treatment. 

### 1.1. Leukemias

Leukemia is a group of blood cancers characterized by the dysfunctional proliferation of tumoral white blood cells. Types of leukemia are classified by the kind of cells affected and by how rapidly the proliferation of tumoral cells occurs (acute or chronic). Among B cell leukemias, we will focus on chronic lymphocytic leukemia and B cell acute lymphoblastic leukemia.

#### 1.1.1. Chronic Lymphocytic Leukemia (CLL)

CLL is a mature B cell neoplasm characterized by the proliferation and accumulation of monoclonal B lymphocytes in PB, BM, lymphoid tissues, and/or extra-nodal areas [1]. It is the most common leukemia in adults in western countries, with an incidence of 4.7 cases per 100,000 inhabitants per year [2]. The median age at diagnosis is 70 years [3]. CLL cells typically exhibit co-expression of the B cell surface antigen CD19, along with CD5, CD23, CD43, and CD200. Compared to normal B cells, tumor cells demonstrate characteristic low levels of surface CD20, surface immunoglobuling (Ig), and CD79b (Table 1) [4]. 

The development of CLL is often preceded by a non-symptomatic precursor state called monoclonal B cell lymphocytosis, defined by a monoclonal B cell count of <5 × 10^9^/L with the typical CLL phenotype, which has a prevalence of up to 12% in aged individuals [1].

Clinical and biological evolution are highly heterogeneous and vary widely among individual patients. Thus, some patients present with an aggressive form of the disease that necessitates treatment shortly after diagnosis and may experience a histological transformation to an aggressive lymphoma—usually diffuse large B cell lymphoma (DLBCL)—commonly referred to as Richter syndrome, while others exhibit an indolent course, characterized by slow disease progression and prolonged survival [3].

Over the past two decades, the prognosis for CLL patients has significantly improved thanks to advancements in therapies. Immunochemotherapy with fludarabine, cyclophosphamide, and rituximab (FCR) was the first regimen to notably enhance overall survival, particularly in younger patients without comorbidities [5]. Subsequently, the CLL10 trial demonstrated that the bendamustine and rituximab (BR) regimen was more active than the FCR and exhibited superior tolerance and comparable survival, thus emerging as a viable alternative for older patients and/or those with comorbidities [6]. In the last decade, the incorporation of new anti-CD20 antibodies (ofatumumab and obinotuzumab), Bruton’s Tyrosine Kinase (BTK) inhibitors (ibrutinib, acalabrutinib, and za-nubrutinib), PI3K inhibitors (idelalisib), and a BCL-2 protein antagonist (venetoclax) has largely replaced immunochemotherapy in many cases due to their superior efficacy (especially in high-risk patients) and tolerability [7,8,9,10,11]. 

#### 1.1.2. Acute Lymphoblastic Leukemia (B-ALL)

Acute lymphoblastic leukemia (ALL) is characterized by the uncontrolled growth of lymphoid cells arrested at an early stage of differentiation, capable of infiltrating the BM, PB, and extramedullary tissues. The incidence is approximately 1.57 cases per 100,000 individuals per year in western countries, with a higher prevalence in children, particularly between 1–5 years. Overall, ALL accounts for about 20% of all childhood cancers in western countries [12]. A BM biopsy may be required to confirm the diagnosis and to determine the presence of blast cells. Immunophenotyping, cytogenetic, and molecular tests are performed to learn the exact type of disease [13]. Pathologic clones can be commonly identified by abnormal expression patterns of classical B cell precursor markers (namely, CD34, CD19, CD10, CD38, and CD20) and, sometimes, also by the acquisition of distinctive aberrant antigens (e.g., CD33 and CD58) (Table 1) [14]. The majority of B-ALL cases are classified according to ploidy changes, such as hyperdiploidy and hypodiploidy, as well as chromosomal rearrangements or the presence of other genetic markers [1]. The main entities are based on the presence of *iAMP21, BCR::ABL1* fusion, *KMT2A* rearrangements, *ETV6::RUNX1* fusion, *TCF3::PBX1* fusion, *TCF3::HLF* fusion, or *IGH::IL3* fusion. B-ALL with *BCR::ABL1*-like features is also an entity which shares the gene expression and phenotypic features of B-ALL with *BCR::ABL1* fusion [15]. Similarly, B-ALL with *ETV6::RUNX1*-like features shares characteristics of B-ALL with *ETV6::RUNX1* fusion [16]. In addition, there is an entity named B-ALL with other defined genetic abnormalities, which includes B-ALL, with other genetic drivers identified by gene expression and sequencing analysis [1]. Treatment options have substantially improved the outcomes for ALL patients, especially among children [17]. Intensive multi-agent chemotherapy has achieved long-term cure in ≥90% of children with ALL, whereas the overall cure rate is around 50% in adult patients. However, this approach is often associated with long-term side effects. Hematopoietic stem cell transplantation (HSCT) can achieve the cure in approximately half of adults, but this approach is also associated with toxicity and therapy-related mortality. In addition, the prognosis of Philadelphia chromosome in (Ph)/BCR/ABL-positive ALL patients has dramatically improved upon the introduction of BCR–ABL tyrosine kinase inhibitors [18]. Immunotherapy with monoclonal antibodies (anti-CD20, anti-CD22, or anti-CD19) has obtained very promising responses and cure rates in B-ALL patients. Moreover, anti-CD19 chimeric antigen receptor (CAR) T cells have shown remarkable efficacy in refractory/relapse B-ALL [19]. Finally, blinatumomab is a CD3/CD19 bispecific T cell-engager antibody that has been recently approved for relapsed or refractory B-ALL patients [20].

### 1.2. B Cell Lymphomas

Lymphomas are a diverse group of hematologic malignancies that can arise from either T cells, B cells, or natural killer (NK) cells. Approximately 450,000 cases of B cell non-Hodgkin’s lymphomas (NHL) are diagnosed annually worldwide, resulting in approximately 240,000 deaths. The development of lymphomas may be caused by alterations that affect normal B cell development, such as chromosomal aberrations—including t(14;18) in follicular lymphoma (FL), t(11;14) in mantle cell lymphoma (MCL), and t(8;14) in Burkitt lymphoma (BL)—dysregulation of signaling pathways, impaired apoptosis/cell cycle regulation, and epigenetic aberrations [21]. 

#### 1.2.1. Diffuse Large B Cell Lymphoma (DLBCL)

DLBCL is the most common subtype of NHL and is known to be an aggressive lymphoma, representing approximately 30–40% of cases with an incidence rate of 7.2 per 100,000 individuals, and involving both nodal and extranodal sites [22,23]. Median age of diagnosis is 70 years [24]. DLBCL represents a heterogeneous group of diseases with variable outcomes that are differentially characterized by clinical features, cell of origin, molecular features, and recurring mutations [25]. It can be classified into two molecular subtypes: activated B cell-like (ABC) and germinal center-derived (GCB), each with distinct signaling pathways [26]. Morphologically, the disease is characterized by diffuse proliferation of large neoplastic B cells with large nucleoli and abundant cytoplasm, which disrupt and efface the underlying architecture of the involved lymph node. These cells typically express pan-B cell antigens, including CD19, CD20, CD22, CD79a, and CD45 (Table 1) [27]. Approximately 14% of cases express CD30, which can result in a favorable prognosis [28]. In terms of treatments, R-CHOP (rituximab, cyclophosphamide, doxorubicin, vincristine, and prednisone) is still the primary form of treatment and has been proven to be successful in long-term disease control in up to 60% of patients at advanced stages and over 90% of patients with limited stages. R-CHOP in combination with novel agents, including ibrutinib, bortezomib, and lenalidomide, has not resulted in better patient outcomes. For patients with chemotherapy-sensitive disease who are transplant-eligible, rescue chemotherapy combined with consolidation of an autologous stem cell transplant (ASCT) is the current standard of treatment. There are now three CAR T cell therapies for refractory or relapsed DLBCL that have obtained FDA approval [29].

#### 1.2.2. Follicular Lymphoma (FL)

FL is the second most frequent subtype of lymphoid malignancy in western Europe (20–30%) following DLBCL, with an incidence rate of 2.2 per 100,000 individuals annually, which is slightly higher among relatives of individuals with FL [30]. The median age at diagnosis is 60 years old [31]. Eventually, the disease spreads from the founder follicle to the surrounding follicles in the lymph node and then to distant lymphoid organs, including the BM, eventually manifesting as a systemic disease [32]. FL cells express monoclonal Ig light chain, CD19, CD20, CD10, and BCL-6, and are negative for CD5 and CD23. In virtually all cases, FL cells overexpress the BCL-2 protein due to t(14;18) (Table 1) [30]. Regarding treatment, in early-stage (I-II) FL with low tumor burden, a balanced approach with localized irradiation and single-agent rituximab may be optimal [33,34]. For asymptomatic advanced stages (III-IV), watchful waiting is appropriate initially, particularly for patients aged >70 years. When treatment becomes necessary and the main objective is complete remission and long progression-free survival (PFS), the optimal course of action is to administer R-CHOP or BR [33]. In a randomized trial, lenalidomide demonstrated non-inferiority to chemoimmunotherapy in initial treatment and, when combined with rituximab, it outperformed rituximab alone in recurrent FL. Kinase inhibitors, SCT, and CAR T therapy are also options for recurrent disease [35].

**Table 1 ijms-25-04644-t001:** List of protein markers used for the diagnosis and classification of each B cell disorder.

	Leukemia		Lymphoma					Myeloma
	**B-ALL** [36,37,38,39,40,41]	**CLL** [41,42,43,44]	**DLBCL** [41,44,45,46,47]	**FL** [41,42,45]	**MCL** [41,42,44,45]	**MZL** [41,44,48]	**BL** [41,45]	**MM** [41,42,49]
**CD3**	−	−	−	−	−	−	−	−
**CD4**	−	−	−	−	−	−	−	−
**CD5**	−	+	−/+	−	+	−	−	−
**CD7**	−	−	−	−	−	−	−	−
**CD8**	−	−	−	−	−	−	−	−
**CD9**	+							
**CD10**	+/−	−	−/+	+/−	−	−	+	−
**CD11c**		+/−	−/+	−/+	−	+	−	−
**CD13**	+/−							
**CD19**	+	+	+	+	+	+	+	−
**CD20**	+	low	+	+	+	+	+	dim+
**CD21**	−							
**CD22**	+	−	+	+	+	+	+	
**CD23**	−	+	−	−	−	−	−/+	+/−
**CD24**	+							
**CD25**		+/−	−	−	−	−/+	−	−
**CD27**		+	+	+	+	+	−/+	−/dim+
**CD28**								+
**CD30**		−	−/+	−	−		−	
**CD33**	+/−							+
**CD34**	+						−	−
**CD38**	+	+/−	−/+	+	+	+/−	+	+
**CD43**		+	−/+	−	+	−/+	+	+/−
**CD44**							low/−	
**CD45**	+	+	+	+	+	+	+	−/+
**CD54**								dim+
**CD56**		−	−					+
**CD58**	+							
**CD66c**	−/+							
**CD73**	−/+							
**CD79a**	+		+	+	+	+	+	−/dim+
**CD79b**		−/low	−/+	+/−	+	+/−	+/−	
**CD81**	+	low/+	+	+	+	+	+	−/dim+
**CD103**		−	−	−	−	−	−	
**CD117**	−							+
**CD123**	+							
**CD138**		−		−	−			+
**CD185**		+	+	+	+	+	+	
**CD200**		+++	−/+	−	−	−	−	+/++
**CD304**	−/+							
**CD305**		−	−	−	−	−	−	
**CD307**								++
**BCL-2**		+	+/−	++	+		−	+/−
**BCL-6**		−	−/+	+	−		+	
**CCND1**					+			
**HLA-DR**	+	+	+	+	+	+	+	
**FCM7**					+			
**Igκ/Igλ**		dim/low	+	+	+	+	+	+
**IgM**	−/+	−	+	−/+	+	−/+	+	+/−
**Ki67**			+				+	
**MPO**	−							
**NG2**	−							
**PAX5**	+						+	
**TdT**	+/−		−				−	

B-ALL, B cell acute lymphoblastic leukemia; BL, Burkitt lymphoma; CD, cluster of differentiation; CLL, chronic lymphocytic leukemia; DLBCL, diffuse large B cell lymphoma; FL, follicular lymphoma; MCL, mantle cell lymphoma; MM, multiple myeloma; MZL, marginal zone lymphoma; +, marker positivity; −, marker negativity; +/−, most of cells are positive for the marker, but negative cells are also present; −/+, most of cells are negative for the marker, but positive cells are also present.

#### 1.2.3. Mantle Cell Lymphoma (MCL)

MCL is a mature B cell neoplasm that accounts for 5% of NHL cases and whose incidence is increasing, being approximately 0.5–1 per 100,000 individuals per year in western countries [50]. The median age at diagnosis is approximately 71 years [51]. The WHO has principally divided MCL into two categories: conventional MCL, with expression of SOX11, unmutated *IGHV*, and an aggressive course; and leukemic non-nodal MCL, characterized by mutated *IGHV*, lack of SOX11, and indolent clinical course [52]. The pathological hallmark of almost all MCL is the overexpression of cyclin D1 (CCND1) [53]. A typical immunophenotypic report by flow cytometry (FCM) and immunohistochemistry (IHC) on tissue biopsy will be positive for CD5, CD20, CD19, FMC7, sIgM/sIgD, CD22, and CD79b, strongly positive for CCND1, and negative for CD23 and CD10 (Table 1) [54]. Rituximab-based chemoimmunotherapy with/without ASCT is the standard first-line treatment for physically fit young patients. Clinical trials have been conducted using BTK inhibitors and BCL-2 antagonists (such as venetoclax) for refractory MCL, showcasing remarkable responses, although resistance to these treatments has frequently been observed. Ongoing investigations into CAR therapy and the synergistic use of non-chemotherapeutic agents signify a shift in emphasis towards rendering MCL treatment devoid of chemotherapy [52,55]. 

#### 1.2.4. Marginal Zone Lymphoma (MZL)

MZL is an indolent B cell lymphoma that accounts for approximately 7% of all B cell NHL cases and has an incidence rate of 1.96 per 100,000 individuals [56]. Median age at diagnosis ranges between 65 and 70 years [57]. MZL includes three different subtypes: extranodal marginal zone lymphoma of the mucosa-associated lymphoid tissue (MALT), nodal (NMZL), and splenic (SMZL) [56]. The development of these diseases can be associated with pathological infections induced by *Helicobacter pylori*, such as gastric MALT and hepatitis C virus (HCV) in all MZL [58]. The differential diagnosis of MZL mainly depends on IHC, including CD20, CD10, CD5, CD23, CCND1, IgD, and SOX11 [59]. In cases of gastric MZL with *H. pylori* infection, eradication of the bacterium is advised regardless of disease stage. Additionally, patients co-infected with chronic HCV should undergo hepatitis treatment and achieve a cure for substantial lymphoma regression. For *H. pylori*-negative gastric MZL, non-gastric extranodal MZL, or cases wherein lymphoma regression does not occur following antibiotic therapy, involved-site radiotherapy has been shown to be highly effective. Symptomatic, advanced-stage MZL is best managed with rituximab-based therapies. In cases of severely symptomatic or bulky disease, combination immunochemotherapy with chlorambucil or bendamustine may be preferred over rituximab alone [60].

#### 1.2.5. Burkitt Lymphoma (BL)

BL is a highly aggressive form of B cell NHL, representing 1–5% of all NHL cases in adults and with an incidence of about 2.5 cases per million per year [61]. BL demonstrates age-dependent incidence rates, with peaks in young children (0–14 years), younger adults (around 40 years), and older adults (age 70) [62]. The disease is associated with Epstein–Barr virus (EBV), human immunodeficiency virus (HIV), and t(8;14) chromosome translocation, which causes overexpression of the *MYC* gene. Three subtypes of BL (sporadic, endemic, and immunodeficiency-associated) can be recognized, with different epidemiology, risk factors, and clinical presentations [63,64]. In terms of diagnosis, the immunophenotypic profiles were CD10^+^, CD19^+^, CD20^+^, BCL-6^+^, Ki-67^+^, TdT^−^, and BCL-2^−^ (Table 1). Additionally, IHC for CD10, CD20, BCL-6, BCL-2, Ki-67, TdT, and MYC translocation is usually performed. Treatment is stratified based on patient age and stage. BL and its variants necessitate specialized treatment protocols, integrating intensive chemotherapy courses containing fractionated alkylating agents and cell cycle phase-specific agents capable of penetrating the blood-brain barrier. Findings from multiple studies indicate that the inclusion of rituximab significantly enhances efficacy, this being particularly beneficial for older patients. Transplantation plays a restricted role, predominantly involving autologous transplants for patients with partial responses to initial therapy or those experiencing chemosensitive relapses [61]. Therapies that inhibit the *MYC* oncogene are currently being investigated. In patients with HIV, highly active antiretroviral therapy has allowed the management of immunodeficiency [64]. CAR T cell therapy targeting CD19 is a promising new treatment option for patients with relapsed or refractory disease. Nevertheless, this therapy is limited by significant toxicity, particularly cytokine release syndrome and neurotoxicity, which restricts its widespread use [65].

### 1.3. Multiple Myeloma (MM)

MM is a hematologic malignancy characterized by the clonal expansion of plasma cells (PC) in the BM, producing a monoclonal Ig or monoclonal component that can be detected in serum and/or urine. MM is the second most common blood cancer behind only NHL, with an estimated incidence of 4–5 cases per 100,000 inhabitants per year in western countries [66]. The median age of the initial diagnosis is 66 years [67,68]. Neoplastic PCs show aberrant phenotypes consisting of multiple possible combinations of the following aberrant antigenic profiles: CD38^+^, CD19^−^, CD45^−^, CD20^+^, CD27^−/low^, CD28^+^, CD56^+/bright^, CD81^−/low^, CD117^+^, and CD200^bright^, in association with monoclonal CyIgκ or CyIgλ [69].

Therapeutic options for MM patients are broad, ranging from polychemotherapy with alkylating agents and steroids to novel agents, such as proteasome inhibitors (PIs; bortezomib, carfilzomib, and ixazomib), immunomodulators (IMiDs; thalidomide, lenalidomide, and pomalidomide), and mAbs (elotuzumab, isatuximab, and daratumumab). Additionally, cell therapy with ASCT and, more recently, novel strategies with bispecific T cell engagers or CAR T cells have been introduced to prolong the survival of high-risk patients [70,71,72]. Standard first-line (induction) therapy consists of a combination of bortezomib, lenalidomide, and dexamethasone, followed by ASCT and maintenance lenalidomide [73,74,75,76].

## 2. Protein-Based Technologies to Study B Cell Malignancies

Protein composition is a dynamic and meaningful parameter to be considered for the characterization of specific cell stages or subtypes. The definition of proteomes provides valuable information for the understanding of molecular and cellular functioning, helping to decipher the tumoral processes and discover potential biomarkers for diagnosis, patient stratification, and prognosis. Such is the case that during the last two decades, tremendous advances have been performed in the clinical field thanks to the investigations performed using protein-based technologies (as further described in Section 3, Table 2, and Figure 2). Next, we describe the basis of the main protein-based strategies employed for studying B cell malignancies.

### 2.1. Mass Spectrometry (MS)

Mass spectrometry (MS) is a powerful analytical technique widely employed in various scientific disciplines that allows for the identification and quantification of peptides, among other molecules, based on their mass-to-charge (*m*/*z*) ratio [144,145]. The origins of MS can be traced back to the early 20th century, although important contributions to the refinement and improvement of mass spectrometers have been made over the years, including high-resolution instruments and novel ionization methods [146]. In this regard, ionization sources are the MS components responsible for converting peptides into ions for subsequent mass analysis. There are three widely used ionization techniques: (i) Surface-Enhanced Laser Desorption/Ionization (SELDI), (ii) Matrix-Assisted Laser Desorption/Ionization (MALDI), and (iii) Electrospray Ionization (ESI). SELDI is a soft ionization technique that combines laser desorption with enhanced ionization via a specialized surface (e.g., a metal-coated chip), where the sample is placed [147]. In MALDI, a peptide sample is embedded in a matrix material, usually a crystalline organic compound that absorbs laser energy. This laser induces desorption and ionization of the peptides [148]. Lastly, ESI involves the generation of droplets of charged ions by applying a high voltage to a peptide-containing liquid sample. 

Likewise, mass analyzers that separate ions based on their *m*/*z* ratios also have a pivotal function in MS. There are various mass analyzers available in the field, although the most employed are: (i) Time-of-Flight (TOF), (ii) quadrupole, and (iii) Orbitrap. In TOF mass analyzers, ions are accelerated through a field of known strength and the time it takes for ions to travel a fixed distance is measured. Lighter ions reach the detector faster than heavier ones [149,150]. Quadrupoles are based on a set of four parallel rods that apply both direct current and radiofrequency voltages. These voltages selectively allow ions of a specific *m*/*z* ratio to pass through to the detector for selective detection in tandem MS. Finally, Orbitrap analyzers trap ions in an electrostatic field, causing them to orbit around a central spindle. The frequency of these orbits is used to determine the mass of the ions, thus obtaining high resolution, mass accuracy, and sensitivity [151,152]. It is also noteworthy that all of these mass analyzers can be used individually or in combination. Hybrid mass spectrometers, incorporating multiple analyzers, are common in modern MS instruments, providing enhanced capabilities for a wide range of applications [153].

MS allows qualitative and quantitative analyses. While the first approach involves the identification of the proteins present in a sample, quantitative analysis focuses on determining the precise amounts of such proteins [154]. In proteomics, innovative techniques have been developed to address quantitative analysis, like iCAT (Isotope-Coded Affinity Tagging) [155], iTRAQ (Isobaric Tags for Relative and Absolute Quantification) [156], SILAC (Stable Isotope Labeling by Amino Acids in Cell Culture) [157], and TMT (Tandem Mass Tags) [158]. iCAT, iTRAQ, and TMT employ chemical tags to label peptides, enabling the simultaneous comparison of protein abundances between different samples. In SILAC, protein labeling is performed during cell culturing. However, MS technology can be affected by several factors, leading to incorrect identifications (a.k.a. false positives) like the presence of contaminants and artefacts in the samples, isobaric interferences, and matrix effects. Isobaric interferences are produced when different compounds with similar *m*/*z* ratios cannot be accurately distinguished from each other with sufficient resolution [159]. While matrix effects refer to ion suppression or enhancement, background noise or signal drift challenges the analyte detection and quantification [160]. Nevertheless, all of these MS techniques have revolutionized the ability to dissect complex biological systems, providing valuable information for research and clinical applications. 

### 2.2. Flow Cytometry (FCM)

Flow cytometry (FCM) is a powerful technique for analyzing and quantifying the characteristics of cells or particles in a fluid suspension [161]. In the conventional method, cells are labeled with fluorescent dyes or antibodies that bind to specific cellular components. As cells pass through a flow cytometer, they are individually interrogated by lasers, and the emitted fluorescence is detected by photomultiplier tubes. The data collected allow for the simultaneous analysis of multiple parameters, such as cell size, granularity, and the expression of specific proteins. A more recent FCM approach is based on the spectral method, where the instrument captures the entire emission spectra of fluorochromes [162]. This allows for improved resolution of closely spaced fluorochromes and reduces spectral overlap, enhancing the ability to distinguish and analyze multiple fluorochromes simultaneously. The spectral method offers greater flexibility in panel design (up to 40 markers) and provides a more accurate and comprehensive understanding of cellular populations.

As occurs with MS, FCM can also present false positives [163]. In this case, cells are incorrectly identified as positive for a particular marker due to factors like non-specific binding of antibodies, a phenomenon that happens when there is cross-reactivity with other cell surface markers. Moreover, the background noise, generally caused by cellular debris, aggregates. Autofluorescent particles and the instrument artefacts derived from electronic noise or light scattering are also sources of false identifications.

### 2.3. Mass Cytometry (CyTOF)

Cytometry by Time of Flight (CyTOF) is a revolutionary technology in the realm of FCM that addresses some of the limitations of traditional fluorescence-based methods. Unlike conventional FCM, CyTOF uses metal isotopes as detection markers instead of fluorochromes [164]. Each metal isotope is conjugated with an antibody, allowing for the simultaneous detection of numerous cellular markers (up to 50 markers). This approach provides higher resolution and the ability to analyze a larger number of parameters in a single experiment, being particularly advantageous for high-dimensional single-cell analysis. As with the other proteomics approaches, CyTOF can present false positives, which arise from non-specific bindings, background noises, and sample artefacts. Additionally, the signal spillover between different metal channels might lead to false identifications. Advances in CyTOF technology have allowed for the reduction of spillover levels to minimum values, reducing the chances of false positives [165]. CyTOF allows for the exploration of complex biological systems with unprecedented detail and depth, and is becoming widely employed in immunology, cancer research, and molecular biology in general, enabling in-depth profiling of immune cell populations and their functional states [166,167].

### 2.4. Other Proteomics Techniques

In terms of molecular biology, another two useful techniques are Western blotting and protein arrays. Western blotting detects and quantifies specific proteins within a complex sample. Proteins are separated by gel electrophoresis, transferred to a membrane, and then probed with specific antibodies that bind to the target proteins. The detection is typically achieved through chemiluminescence or fluorescence, allowing for the visualization and quantification of the proteins of interest [168]. Protein arrays involve the immobilization of many proteins on a solid surface, allowing for the simultaneous study of multiple interactions in a single experiment. Protein arrays are particularly valuable for high-throughput screening of protein–protein interactions, identification of binding partners, and profiling of protein expression patterns [169].

### 2.5. Integration of Proteomics with Other-Omics Approaches

The term “-omics” encompasses various high-throughput approaches that enable the analysis of large-scale biological components, such as genes (genomics), transcripts (transcriptomics), proteins (proteomics), and metabolites (metabolomics) [170]. Notably, proteomics studies could be beneficial because proteins are the critical effectors of cell functions and phenotypes, and they could better obtain the dynamic properties of cells. In addition, genomics and transcriptomics data alone cannot show a full landscape of the disease mechanisms with a comprehensive understanding, since they could fail to reveal post-translational modifications and protein degradation and, therefore, not accurately reflect the protein levels [171]. In fact, over the last years, phosphoproteomics approaches have experienced explosive growth that have gained insight into the biological and clinical areas of cancer [172]. One of the challenges in this field is data integration of different “-omics”, since a single platform is insufficient to decipher the complexity underlying cancer cells. In this regard, the integration of multi-omics approaches unveils not only the information of a single cellular layer but also the relationships between layers, enabling the establishment of multi-dimensional molecular landscapes and elucidating the flow of information, from the original cause to the functional consequences [173].

## 3. Proteomics Studies for the Understanding of B Cell Hematological Malignancies

Proteomic studies play a critical role in unraveling the complexities of hematologic B cell malignancies. Understanding the diverse protein profiles, modifications, and interactions within malignant B cells may offer valuable insights into the underlying mechanisms driving these cancers and allow for the identification of specific protein biomarkers associated with disease diagnosis, progression, outcome, and response to therapy. Below, we discuss the latest contributions of proteomic studies to the diagnosis, prognosis, and treatment of these diseases (Table 2). Furthermore, Figure 3 displays the number of manuscripts published per year (2018–2023 period) related to the application of FCM, MS, and mass cytometry for the study of CLL, MM, and lymphomas.

### 3.1. Proteomics Studies on Chronic Lymphocytic Leukemia

Understanding this complex and heterogeneous disease is challenging, and there still remain gaps in the knowledge of mutational-related tumor phenotypes, mainly due to the lack of comprehensive proteomic characterizations. Meier-Abt et al. [77] profiled 117 CLL samples by MS and integrated the data with other omics approaches (genomics and transcriptomics) and drug response assays, providing the proteome landscape of CLL. With >3300 identified proteins, it was shown that trisomy 12 and *IGHV* mutational status were the determining parameters of CLL variation at the protein level. Moreover, the BCR/PI3K/AKT cell signaling pathway was confirmed to be key for tumorigenesis.

The largest study cohort involving 795 CLL (and other related leukemias and healthy) cases was screened by reverse phase protein array (RRPA) (https://www.leukemiaatlas.org/, accessed on 1 April 2024) [140]. Specifically, the expression of 384 different proteins (80 of them including post-translational modifications linked to activation status) was evaluated for prognostic and therapeutic purposes. In summary, six proteomic signatures were defined to predict survival in CLL, including one group showing resistance to all available therapies. Likewise, a CLL-group classification was made by analyzing the proteome profiles of 68 patients using in-depth high-resolution isoelectric focusing liquid chromatography (LC)-MS [78]. In this multi-omics study, integrating genetic alterations, mRNA expression, and protein abundance levels, a new CLL subgroup characterized by high expression levels of spliceosomal proteins was revealed and linked to poor clinical outcomes.

In addition to MS- and array-based proteomics investigations, FCM approaches are routinely used in the clinical setting to diagnose and perform the follow-up for CLL patients. The knowledge of specific CLL B cell-related markers allows for the design of FCM panels, which provide clinicians with quick results on disease diagnosis and progression. Such is the case of the EuroFlow consortium (https://euroflow.org/, accessed on 1 April 2024), which has defined combinations like the EuroFlow^®^ Lymphocyte Screening Tube (LST; eight-color tube for the identification of the major leukocyte populations in PB) or the EuroFlow^®^ Ig isotype subclass B cell tube (twelve-color tube for the identification of the different subsets of normal residual PB B cells based on the expression of distinct Ig subclasses). New advances in the field (spectral FCM) have permitted the inclusion of extra markers to further characterize the samples. In addition to clinical applications, FCM has also been applied in research to, for instance, compare CLL vs. MBL [113].

Finally, state-of-the-art mass cytometry technology offers the possibility of high-dimensional screenings. The first CyTOF study on CLL dated from 2018, when a 35-marker panel was designed to dissect the tumor microenvironment (TME) of a malignancy [127]. Key protein players like PD-L1, LAG3, CD144, and ICAM-1 were more expressed in tumoral cells compared to the normal B cells, and researchers also detected heterogeneous expression of PD-L1 within the CLL group. In another case, CyTOF technology was used to develop a 24-protein panel to analyze a healthy B cell pool of CLL patients, revealing low numbers of anergic self-reactive B cells in those compartments and suggesting that CLL might be linked to an abnormal B cell lymphopoiesis [128].

### 3.2. Proteomics Studies on Acute Lymphoblastic Leukemia (B-ALL)

Proteomics approaches have shown promise in ALL, with potential applications in identifying disease mechanisms, prognostic biomarkers, and new target therapies.

Regarding prognosis, Braoudaki et al. utilized MALDI-TOF MS to identify potential prognostic biomarkers. This study highlighted several proteins, such as CLUS, CERU, and APOE, which may serve as distinctive biomarkers for leukemia aggressiveness or therapeutic targets in pediatric ALL [79]. Additionally, Yu et al. utilized MS to explore potential biomarkers and autoantigens linked to childhood B-ALL, identifying α-enolase and VDAC1 as promising serological markers for diagnosis and prognosis. Immunohistochemical analysis further revealed positive expression of α-enolase and VDAC1 in 95% and 85% of B-ALL patients, respectively [85]. Another study employed nano-UPLC tandem nano-ESI-MS(E) to analyze serum proteomes in pediatric B-ALL patients, identifying candidate biomarker proteins for early diagnosis and treatment evaluation. This study reported several upregulated proteins in B-ALL patients at diagnosis (LRG1, CLU, F2, SERPIND1, A2M, SERPINF2, SERPINA1, CFB, and C3) [81]. Furthermore, using FMC, Abaza et al. suggested that NRP-1/CD304 expression could serve as a reliable tool for distinguishing leukemic B-lymphoblasts from normal counterparts, offering a stable phenotype for minimal residual disease monitoring and potentially indicating poor prognosis in pediatric precursor B-ALL [114].

Regarding response treatments, Dehghan-Nayeri et al. identified three differentially promising biomarkers for prognosis and chemotherapy response in childhood ALL (VDAC1, SNX3, and PFDN6). These proteins play crucial roles in drug resistance, apoptosis regulation (VDAC1), protein trafficking (SNX3), and protein folding (PFDN6). Moreover, their expression was downregulated in high-risk ALL cases. Protein-–protein interaction analysis revealed their involvement in the NR3C1 signaling pathway, the target of dexamethasone [82]. To understand the chemoresistance to vincristine, Guzmán-Ortiz et al. used nanoHPLC coupled with an ESI-ion trap mass to identify 135 proteins that were exclusively expressed in the presence of vincristine. The most represented functional categories were Toll receptor signaling pathway, Ras pathway, B and T cell activation, CCKR signaling map, cytokine-mediated signaling pathway, and oxidative phosphorylation. This study indicated that signal transduction and mitochondrial ATP production are essential during the adaptation of leukemic cells to vincristine treatment. These processes represent potential therapeutic targets for intervention in B-lineage pediatric ALL treatment [83]. Additionally, it is crucial to biologically understand why treatments may not be effective in some patients. In this context, the integration of genomics and proteomics approaches showed that IKZF1 perturbation plays a central role in BCR-ABL1 lymphoid leukemogenesis and has a reduced response to tyrosine kinase inhibitor (TKI) therapy [80]. Despite new advances, there are still subsets of ALL patients with poor response who relapse. To address this issue, a recent study conducted a comprehensive multi-omic analysis of proteomics, transcriptomics, and pharmacoproteomics of 49 readily available childhood ALL cell lines, aiming to elucidate the molecular profiles of these cell lines and their association with drug responses to 528 oncology drugs. Through this analysis, the researchers identified correlations between specific drugs and molecular phenotypes, as well as lineage-dependent correlations [84].

### 3.3. Proteomics Studies on Diffuse Large B Cell Lymphoma (DLBCL)

It is well known that the interplay between tumoral B cells and T cells influence disease development via the regulation of B lymphoma cells. In this regard, Rusak et al. [115] developed a marker panel for B cells (CD19^+^, CD20^+^, CD22^+^, and CD79a^+^), T cells (CD3^+^/CD4^+^/CD5^+^/CD8^+^), and Treg cells (CD4^+^/CD25^+++^/Foxp3^high^) to determine their absolute counts and correlate such information for the prognostication of newly diagnosed DLBCL patients. They have also suggested the usage of monocytic population distribution in DLBCL patients as an independent prognostic factor [116]. Thus, a greater number of classical (cMo, CD14^+^/CD16^−^) and intermediate monocytes (iMo, CD14^+^/CD16^+^), and a decrease in the non-classical counterpart (ncMo, CD14^low^/CD16^+^) were detected in the PB of patients with favorable prognosis. This investigation was performed using a seven-color FCM panel, demonstrating once again the applicability of this technique in routine clinical practice. Moreover, it was shown that the increase in ncMo was linked to an adverse prognosis. Other myeloid cells not extensively investigated in DLBCL disease are the tumor-associated macrophages (TAM). Using mass cytometry, Ferrant et al. [130] reported a co-regulation between TAMs and the activation status of blood endothelial cells in lymphoma via an alteration of the annexin A1 and the formyl-peptide receptor axis.

A broader immune characterization was performed using a 42-marker CyTOF panel, describing the relative frequencies of major immune subsets (i.e., B cells, T cells, and monocytes) [132]. In this study, the authors aimed to find differences between DLBCL and DHL (double-hit lymphoma), an aggressive high-grade B cell lymphoma presenting rearrangements of *MYC* and *BCL2* genes. Their investigation reported an enrichment in B cells and monocytes in DLBCL vs. DHL patients, whereas NK cells and T cells were diminished in the former malignancy. Within T cells, similar levels of CD8^+^ T cells, absence of double-positive T cells, higher numbers of CD4^+^ T cells, and lower levels of γδ T cells were observed in DLBCL compared to DHL samples. As for the B cell compartment, DLBCL patients showed an increase in plasma and memory B cells and a decrease in naïve B cells. Almost no granulocytes were detected and the classical and intermediate monocytic subpopulations were enriched.

To distinguish between DLBCL and FL, another FCM approach for the study of the cell cycle was applied [117]. Here, the detection of high rates of DNA aneuploidy and DNA indexes allowed for the identification of DLBCL vs. FL. DLBCL samples showed a more proliferative profile with a greater number of cells in the S phase. Although this strategy on its own might still require other parameters to confirm the diagnosis, it is of great relevance when biopsies are not available.

To examine tumor heterogeneity at the protein level, a mass cytometry panel of 38 superficial and intracellular markers was designed [129]. Results reported high levels of intra- and inter-variability, resulting in the presence of multiple tumoral clones within each individual (up to 44 different population clusters). Normal vs. abnormal classification was made based on the expression of specific markers (e.g., kappa/lambda light chain). In addition, the integration of proteomics and genomics data in this study allowed for the correlation of phenotypically different cell subpopulations with genetically distinct subclones. In other cases, mass cytometry is used as a validation strategy. For instance, Shi et al. [131] performed a single-cell RNA-seq analysis on relapsed and refractory DLBCL patients using PB mononuclear cell samples, identifying 35 unique markers expressed in these patients vs. healthy donors. Based on this protein panel, mass cytometry was applied in an extra cohort of patients, revealing the importance of 12 of those proteins (CD14, CD31, CD36, CD55, CD59, CD63, CD69, CD82, CD84, CD163, CD226, and IKZF1) which were significantly overexpressed in the DLBCL samples.

For a deeper characterization, LC-MS strategies are commonly used. Thus, van der Meeren et al. [87] profiled the two subtypes of DLBCL (GCB and non-GCB), analyzing more than 4200 proteins of which 37 were found significantly differentially expressed between the two subtypes. Validation of these markers revealed that the expression of glomulin protein was higher in GCB-DLBCL patients, whereas ribosomal protein L23 was a biomarker for non-GCB-DLBCL individuals. Similarly, Ednersson and collaborators [86] reported the enrichment of pathways related to the immune system, interferon signaling, antigen processing, and down-modulation of cell surface receptors in the non-GCB subtype, also applying MS approaches. Another study on the same topic explored the proteome of extracellular vesicles (EVs) reporting >200 proteins differentially expressed between GCB and non-GCB subtypes [124].

### 3.4. Proteomics Studies on Follicular Lymphoma (FL)

Improved methods in MS have opened new approaches to identify biomarkers for early disease detection and management of FL. A novel LC-MS/MS and ‘Total Protein Approach’ analysis identified more than 1000 differentially abundant proteins between lymphoma and control samples. Dysregulated proteins were enriched in biological processes such as the BCR signaling pathway, cellular adhesion molecules pathway, or membrane trafficking [93].

Proteomic analyses have also revealed potential predictive indicators of histological transformation (HT) in FL [94]. Recently, Label-Free Quantification (LFQ) nLC-MS/MS analysis has shown differential protein expression between FL samples from patients with subsequent HT compared to patients without HT, revealing protein patterns associated with a high risk of HT. Notably, IHC analysis confirmed higher expression levels of apoptotic proteins such as CASP3, MCL1, BAX, BCL-xL, and BCL-Rambo in transforming FL [94].

Proteomic studies have also provided valuable understanding regarding treatment response in FL. A study utilizing iCAT-LC-MS/MS identified differentially expressed proteins after p38 MAPK inhibitor treatment, most of them downregulated. These proteins had a role in overlapping pathways, such as IL-6/PI3K, IGF-2/Ras/Raf, WNT8d/Frizzled, MAPKAPK2, and NF-κB pathways, providing insights into treatment effects on FL cells [89]. Another study focused on the understanding of nucleoside analogue resistance by the application of a new LC-MS/MS method to quantify intracellular araCTP (1-beta-D-arabinofuranosylcytosine triphosphate), CTP (cytidine triphosphate), and dCTP (deoxycytidine triphosphate) in human cell extracts [91].

Protein expression profiling of FL vs. MCL using MS analysis (MALDI-TOF) showed 38 differentially expressed proteins related to DNA repair, cell cycle control, transcription, and apoptosis. Furthermore, the comparison of the proteome with the RNA expression array data revealed only a modest correlation between RNA and protein, which emphasized the relevance of post-translational regulation in lymphomagenesis [95].

### 3.5. Proteomics Studies on Mantle Cell Lymphoma (MCL)

Significant advances have been achieved thanks to proteomic techniques in MCL. SELDI-TOF-MS identified distinct molecular signatures that differentiated MCL from B cells of the different compartments (the germinal center, GC, the pre-GC mantle zone, and the post-GC marginal zone), including several upregulated transmembrane proteins, such as CD27, CD70, and CD31 [96]. Moreover, CD148 has been proposed as a candidate proteomic biomarker for distinguishing MCL using nano LC-MS/MS [97]. Lastly, a study used SELDI-TOF-MS to identify specific proteomic biomarkers for MCL, which are histones H2B and H4, overexpressed in MCL tumor biopsies [98].

Proteomics together with other omics are valuable techniques for the development of new therapeutic targets in MCL. One study integrating genomic and proteomic approaches, such as LC-MS/MS, has uncovered neo-antigens in MCL. A systematic characterization of major histocompatibility complex ligands from 17 patients revealed that all neo-antigenic peptides found originated exclusively from lymphoma Ig heavy or light chain variable regions. Moreover, the study allowed for the isolation of circulating CD4^+^ T cells specific for Ig-derived neo-antigens, which could mediate the killing of autologous lymphoma cells [101]. Another study has shown the impact of p53 activation induced by the MDM2 antagonist, nutlin-3a, by an integrative analysis of transcriptomics and proteomics using nLC ESI-MS/MS and other techniques. This study revealed a system-wide effect of nutlin-3a on 4037 differentially affected proteins, uncovering diverse pathways perturbed by p53 activation (PI3K/mTOR pathway, heat-shock response, glycolysis, etc.) Moreover, the identification of synergistic apoptotic effects upon the combined inhibition of the HSP90 or PI3K/mTOR pathway with nutlin-3a-induced p53 activation suggested a promising therapeutic strategy against lymphomas [102].

Proteomic techniques also play a crucial role in advancing our understanding of the effects of different treatments on MCL. In the context of relapsed MCL, one study examined the significance of monitoring patient serum proteomes during triple combination therapy with rituximab, ibrutinib, and lenalidomide. Researchers employed microarray protein technology to identify proteins modulated by treatment, particularly in MCL with ATM/TP53 alterations. Specific proteins, such as TGF-β1, CD40, and complement component 4, were linked to the presence of minimal residual disease in the treated samples. Elevated BTK levels were also associated with shorter PFS [142]. In another study, LC-MS/MS analysis of salivary samples before, during, and after chemotherapy provided valuable insights into the dynamics of the disease and response to treatment [99].

### 3.6. Proteomics Studies on Marginal Zone Lymphoma (MZL)

Having reliable biomarkers is crucial for early detection and the development of less invasive diagnostic approaches. This motivated Cui et al. [104] to explore Primary Sjögren’s syndrome (pSS), a condition that can lead to the development of MALT. They identified four overexpressed proteins (cofilin-1, alpha-enolase, annexin A2, and Rho GDP-dissociation inhibitor 2) in pSS and pSS/MALT cells using 2D gel electrophoresis/MS. Moreover, the expression levels of salivary anti-cofilin-1, anti-alpha-enolase, and anti-RGI2 were found to be higher in pSS/MALT patients than in healthy controls, indicating their potential as biomarkers for diagnosis and prediction of progression.

Understanding the pathogenesis of MZL is critical, and proteomic studies have shed light on the intricate profiles of various MZL subtypes. *Chlamydophila psittaci*-negative ocular adnexa extra-nodal marginal zone lymphoma (OAEMZL), a subtype of MALT, expresses a biased repertoire of mutated surface Ig. Studies using microcapillary reverse-phase high-performance liquid chromatography (HPLC)-MS/MS have confirmed several reactive antigens, including shared intracellular and extracellular self-antigens like galectin-3. Notably, these self-antigens induced BCR signaling in B cells expressing surface Ig derived from OAEMZL. This suggests a functional relationship between self-antigens and BCR-derived tumors, shedding light on potential mechanisms in the pathogenesis of OAEMZL [103].

A study analyzed FCM results to distinguish ocular adnexa lymphoproliferative disorders such as IgG4-related ophthalmic disease (IgG4-ROD), idiopathic orbital inflammation (IOI), and OAEMZL. Results showed significant differences in the expression of CD2, CD3, CD4, CD7, and CD10 between IgG4-ROD/IOI and OAEMZL patients [119].

### 3.7. Proteomics Studies on Burkitt Lymphoma (BL)

Despite comprehensive gene expression studies carried out on BL, the proteomic signatures of this disease remain unexplored. One study used iTRAQ analysis to identify differentially expressed proteins in BL. Over-expression of proteins linked to lymphomagenesis, such as TUBB2C, UCHL1, and HSP90AB1, was found. The endemic (eBL) and sporadic (sBL) variants had distinct protein expression patterns, with C1QBP and ENO1 over-expressed in eBL, and PCNA and SLC3A2 in sBL. DDX3X was over-expressed in EBV^+^ BL cell lines and B cells compared to EBV^−^ cells [107]. Following the lymphomagenesis research, a recent study using MS found that ID3, which is one of the most downregulated genes in BL, had a role in regulating cell proliferation through deregulation of TCF3 and TCF4 in BL cells [105].

### 3.8. Proteomics Studies on Multiple Myeloma (MM)

Proteomic studies have demonstrated numerous applications in MM. CyTOF analysis of PB cells across the developmental stages of myeloma (MGUS, SMM, MM) and in healthy individuals uncovered important differences in the immune cell compartments, including significantly diminished B cell frequencies in total leukocytes and reduced numbers of Treg cells in MM patients compared to their healthy counterparts [133]. Another study using CyTOF profiled PC and maturation stages of B lineage differentiation during MM evolution to characterize the heterogeneity of the disease. Overexpression of MMSET, Notch-1, and CD47, along with variations in B cell signaling regulators (IRF-4, Bcl-6, c-Myc, CXCR4, MYD88, and spliced XBP-1), PC aberrant markers (CD319, CD269, CD200, CD117, CD56, and CD28), and stemness-controlling markers were associated with different clinical outcomes [137]. Baughn and collaborators also identified common patterns of subclonal protein profiles that correlated with clinical behavior. One cluster characterized by increased CD45 and reduced BCL-2 expression was associated with favorable treatment response and improved overall survival. These associations were independent of tumor genetic abnormalities, highlighting the potential of protein profiling as a powerful tool for prognosis and treatment stratification in MM [139].

Various studies have demonstrated the role of T cells and innate immune populations in MM pathogenesis [174]. A comprehensive investigation of the proteome of primary T cell subtypes in the PB of MM patients was conducted using a combination of FCM and the simple integrated spintip-based proteomics technology (SISPROT). This approach enabled the characterization of global proteomes for CD3^+^, CD4^+^, and CD8^+^ T cells, as well as subtype-specific proteomes for eight distinct T cell subtypes. Moreover, a two-step machine-learning-based subtyping strategy was developed and successfully applied to classify T cell subtypes. This innovative approach facilitated the efficient extraction of unique proteome classifiers from PB samples of MM patients [126].

The interplay between the immune TME (iTME) and the malignant clone plays a crucial role in either promoting or inhibiting PC growth. In a study led by Kourelis et al. [134], a 33-marker CyTOF panel was employed to characterize the iTME in patients with MGUS and MM at diagnosis and post-initial therapies. The investigation revealed novel phenotypes with potential roles in tumor immunosurveillance and immune escape. Distinct features of T cell exhaustion were identified in certain CCR5-expressing T cell subsets, suggesting a potential inability to control tumor growth, and implying that T cell immune senescence might contribute to disease progression after therapy. The early post-ASCT period exhibited significant immunosuppression, marked by an increase in CD38^+^ Tregs, which may suggest the incorporation of anti-CD38 therapies in the maintenance setting.

Another study by Dhodapkar et al. [175] focused on the TME to identify determinants of durable responses to BCMA CAR T therapy by utilizing a comprehensive approach combining several high-dimensional single-cell techniques, including CITE-seq (cellular indexing of transcriptomes and epitopes by sequencing), single-cell transcriptomics, mass cytometry, and T cell receptor (TCR) sequencing. The analysis revealed that a lower diversity of pre-therapy TCR repertoire, the presence of hyperexpanded clones exhibiting an exhaustion phenotype, and cells with less differentiated phenotypes (BAFF^+^PD-L1^+^) correlated with shorter PFS following CAR T therapy. This suggests that preserving or restoring TCR diversity during therapy may be a critical factor in achieving durable immune control with T cell-based immunotherapy.

The functional cytotoxicity of cells plays an increasingly vital role in controlling MM, prompting a growing focus on NK cells for their potential to recognize and eliminate malignant PC. In a study conducted by Seymour et al. [136], mass cytometry was utilized to characterize NK cells in newly diagnosed MM cases, revealing features indicative of NK cell exhaustion, affecting both NK CD56^bright^ and NK CD56^dim^ populations, which were associated with a survival disadvantage. The restoration of NK cell function through immune-targeted therapies may be a promising avenue by which to reinstate immune control in MM. Conversely, depleting myeloid-derived suppressor cells (MDSCs) stands out as a potentially crucial strategy to enhance and prolong the effectiveness of novel immunotherapies such as CAR T cells or T cell engager bispecific antibodies. MDSCs play a role in promoting tumor growth and inducing immune suppression. In a comprehensive approach that integrated clinical, functional, and molecular data on granulocytic cells from the TME, Pérez et al. [123] identified a set of markers (CD11b/CD13/CD16) for optimal monitoring of granulocytic MDSCs in MM that was evaluated in BM samples from both controls and MM patients using multi-dimensional FCM. An RNAseq of immature, intermediate, and mature neutrophils was performed to validate CD11b, CD13, and CD16 as robust markers to identify and isolate neutrophil stages. Additionally, the study revealed that a high mature neutrophil/T cell ratio resulted in inferior PFS.

An investigation employing mass cytometry to analyze the immune checkpoint signature in MM identified several critical immune checkpoints that may serve as novel targets for the development of potent checkpoint-blockade-based MM immunotherapeutic strategies. The interaction between MM cells and surrounding immune cells led to immune checkpoint dysregulation. Immune checkpoint ligands such as GAL9, ICOSL, HLA-DR, CD86, PD-L2, and 4-1BBL were predominantly presented on MM cells, influencing the immune response through binding to their receptors on immune effector cells [135]. The PD-1/PD-L1 axis, a central immune checkpoint controlling antitumor responses, was studied by Costa et al. in the BM niche of patients with MGUS, SMM, and active disease. FCM analysis revealed no significant differences in the PD-L1/PD-1 immune profile between patients with SMM and those with active MM. However, PD-L1 expression by CD138^+^ cells was higher in MM compared to MGUS patients. Patients who experienced relapse exhibited an inverted CD4^+^/CD8^+^ ratio, elevated levels of pro-tumoral IL-6, and a positive correlation between %CD14^+^PD-L1^+^ and %CD8^+^PD-1^+^ cells. This suggests a highly compromised immune compartment with a low number of CD4^+^ effector cells. In contrast, SMM patients demonstrated a less compromised and more responsive immune compartment with a normal CD4^+^/CD8^+^ ratio, indicating that SMM patients could be promising candidates for PD-L1/PD-1 inhibition therapy [176].

In the context of identifying biomarkers that can predict progression in individuals with MGUS, Dispenzieri et al. demonstrated that the risk of progression to PC disorders was higher among patients with N-glycosylated Ig light chains (LCs) by utilizing MALDI-TOF-MS. This discovery implies the possibility of diagnosing these diseases earlier and supports the recommendation that MGUS patients with N-glycosylated LCs should be closely monitored as high-risk for MGUS [108].

The development of new therapeutic drugs remains a focal point in MM research. A recent approach developed by Ferguson et al. involved glycoprotein cell surface capture (CSC) combined with LC-MS/MS, allowing for the direct quantification of hundreds of surface proteins on myeloma tumor cells. The proteomic data was integrated with RNA-based datasets to identify potential immunotherapy targets and biomarker candidates associated with resistance and response to treatment. CCR10 emerged as a promising target, being widely expressed on malignant PC. This discovery was validated by testing CAR T cells engineered with their natural ligand, CCL27, showcasing the potential of this approach in advancing targeted therapies for MM [109]. In the same line, Di Meo et al. introduced another method called Immune-TargetFinder, designed to identify biologically and therapeutically relevant cell surface targets for engineered T cells and antibody development in MM. This approach integrated MS data from MM cell lines with RNA-seq data from MM patients. Noteworthy targets identified included ILT3, SEMA4A, CCR1, IL12RB1, FCRL3, and LRRC8D. A bispecific T cell engager targeting ILT3 in MM cells was developed, demonstrating potent killing effects [110].

In the context of personalized MM therapy, the identification of distinct immune cell phenotypes is relevant for predicting treatment response. An FCM analysis of BM and PB samples from patients undergoing daratumumab treatment revealed that the response to daratumumab monotherapy is, in part, dependent on the baseline CD38 expression levels in tumor cells, and that treatment leads to a significant reduction in CD38 expression independent of treatment response, indicating that it is not the sole mechanism of daratumumab resistance. CyTOF profiling of immune cell subpopulations confirmed the decreased CD38 expression induced by daratumumab. Not only are CD38^+^ basophils and NK cells (CD45^+^CD3^−^CD56^+/dim^) reduced, but immune-suppressive cells also show a decrease in patients treated with daratumumab. Furthermore, there was an increased expression of activation markers, such as CD69 and CD127, in the remaining NK cells, suggesting an adaptive response that could contribute to the depth of response [120,121,122]. Similarly, multi-scale data from high-dimensional phenotyping provided valuable biological and mechanistic insights into patients receiving the anti-PD-L1 monoclonal antibody atezolizumab (Atezo). Baseline CyTOF data revealed that patients in the refractory cluster had higher numbers of senescent-type CD57^+^ T cells, which were characterized by their inability to inhibit the growth of malignant cells, providing important information for understanding and potentially overcoming treatment resistance [138].

### 3.9. Case Studies Using Proteomics

Proteomics has the potential to improve the management of patients with B cell malignancies. For instance, FCM has proven to be very useful in the diagnosis of leukemias and lymphomas with central nervous system involvement, serving as a less invasive alternative to cerebral biopsies, which carry a high risk of severe complications. In two clinical cases presented here, cerebrospinal fluid FCM facilitated accurate diagnosis, enabling clinicians to initiate appropriate therapies [177,178]. The high sensitivity of this technique allows for its application even with limited samples, particularly in challenging scenarios where accessing lymph nodes is impractical. This was demonstrated in another case where FCM immunophenotyping of ascitic fluid detected a DHL in a relapsed DLBCL patient [179]. Moreover, evaluating early therapy response via FCM proved valuable in predicting outcome and optimizing the treatment strategy for a patient with primary BM B cell lymphoma [180].

MS-based assays can be used to monitor changes in protein expression levels throughout treatment. Clinical cases have reported a correlation between salivary proteomic profiles and clinical responses in conditions like pSS, MALT-type parotid lymphoma, and MCL, indicating the potential use of saliva samples for monitoring disease progression and treatment efficacy [99,181]. Another clinical application of MS is in diagnosing infectious diseases in patients receiving chemotherapy. This facilitates treatment adjustments and reduces complications, hospital stays, and mortality rates, as observed in reported cases [182,183,184,185].

## 4. Conclusions and Future Perspectives

B cell malignancies comprise a heterogeneous group of diseases originating at different stages of B cell differentiation. Since their discovery, many investigations have been performed aiming to decipher the molecular and cellular mechanisms behind their development. Such information helps to better select targeted therapies to handle these diseases. Therefore, the application of multiple analytical approaches to profile these cancers is highly beneficial. In this regard, the characterization of the protein turnover within leukemias, lymphomas, and myelomas is critical and offers an accurate picture of each cell state, since proteins are the final molecular effectors. Thus, FCM, MS, and, more recently, mass cytometry deliver thorough descriptions of proteomes.

In general, these proteomics techniques are complex and costly and require trained personnel and specialized informatics knowledge for data integration. Despite the fact that FCM has historically been used daily in clinics for the diagnosis and follow-up of patients suffering from any B cell malignancy, only the EuroFlow consortium offers standardized guidelines for the diagnosis and classification of hematological malignancies. Its main goal is to improve the quality of diagnostics in the field of hemato-oncology worldwide. However, not all hospitals and laboratories follow their recommendations, hampering data comparison across the world. In terms of flexibility for marker identification, FCM allows now for up to 40 markers to be simultaneously detected. This has resulted in a deeper characterization of cellular components and their frequencies in a quick manner, providing clinicians with accurate results in a few hours after sample uptake. In research, FCM is also leading the proteomics-related publication ranking for B cell tumors (Figure 3A). The observed trend indicates a greater usage of this technique for lymphomas, followed by MM and CLL (Figure 3B), most likely due to the higher heterogeneity within the lymphoma group, with many different malignancies included. Despite its benefits, FCM is a targeted technique to identify the cells of interest, which is ideal for patient diagnosis and/or follow-up but presents limitations for further unbiased findings. Moreover, the discovery of novel subpopulations within the major B cell subsets, showing closely related profiles, increases the analysis complexity, requiring the inclusion of more markers (or more marker panels) for reliable cell discrimination. In this regard, mass cytometry works similarly to FCM but includes more markers for cell profiling (>50). Compared to FCM, the design of marker panels for mass cytometry is simpler and the usage of metal tags (instead of fluorochromes) eliminates autofluorescence-derived issues. Nevertheless, measurement throughput is much lower in mass cytometry than in FCM (113 min vs. 6 min for 10^6^ cells, respectively) [186]. This challenges the application of this new technique in daily clinical practice, and it is currently used mainly for research purposes (up to 60 publications on B cell malignancies in 2023, Figure 3B). Furthermore, the recent development of the mass cytometry technique results in the lack of any standardized guidelines for sample and data processing, hindering its clinical application since no consensus is available. Nevertheless, the upgrades made in the mass cytometer instrument to allow for automation of sample acquisition might facilitate its application in routine treatment. Samples can be acquired during the night and analyzed the next morning, providing the clinician with an analysis report within 24 h from patient sample collection. Additionally, barcoding in mass cytometry allows for pooled samples from different donors to be analyzed within one single measurement, further reducing the acquisition times. MS appears as a good alternative to screening samples in a high-throughput manner by reporting and quantifying thousands of proteins in an unbiased way. However, its application in clinics is limited due to the long sample processing and sample acquisition procedures, together with the requirement of highly experienced users. Yet MS investigations often lack a deep analysis of the obtained results referring to cellular and molecular mechanisms, only reporting protein lists. Moreover, occasionally, when comparing studies, contradictory data can be found which point out the need for meta-analyses. Nevertheless, MS is considered a valuable approach for research purposes, since it enables comprehensive profiling using low sample amounts. During the last few years, it has been widely applied for MM studies, with an increased interest in recent years (Figure 3B). Finally, it is noteworthy that single-cell proteomics is emerging as a promising technology for the characterization of individual cells. As it occurs with FCM, CyTOF, and single-cell transcriptomics approaches, single-cell analysis is critical to reveal population heterogeneity (allowing for identification of minor subpopulations) and to uncover unique characteristics of individual cells. Different attempts have been made to achieve this proteomics characterization at single-cell level, like the SCoPE-MS technology [187], although further improvements are still required in the field.

In summary, protein-based approaches can be considered as suitable strategies for the characterization of B cell malignancies, in both targeted and unbiased ways. Different options are available for clinical and research applications, and the continuous progress in the field guarantees their potential for deeper analyses. Furthermore, the integration of proteomics data with other -omics-derived information (e.g., single-cell transcriptomics, metabolomics) might benefit the comprehensive understanding of these diseases and the development of personalized medicine in B cell disorders.

## Figures and Tables

**Figure 1 ijms-25-04644-f001:**
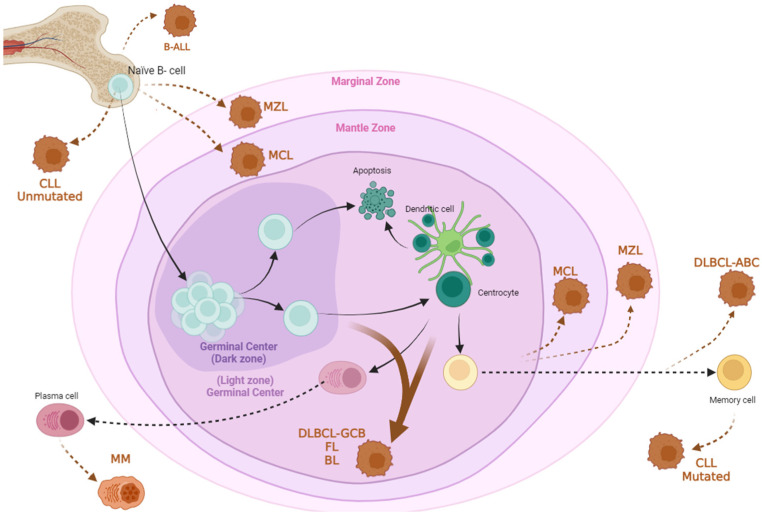
**Cellular origins of the main subtypes of human B cell hematological malignancies.** Mantle cell lymphomas (MCL) and marginal zone lymphomas (MZL) can be transformed from either naïve B cells or germinal center-derived cells. Follicular lymphomas (FL), Burkitt lymphoma (BL), and diffuse large B cell lymphomas (DLBCL-GCB) originate from B cells within the germinal center. DLBCL-ABC appears to originate from B cells within the marginal zone or from fully differentiated memory B cells. Chronic lymphocytic leukemia (CLL) cells can transform either before or after IGHV rearrangement. B cell acute lymphoblastic leukemia (B-ALL) originates from naïve cells. Finally, multiple myeloma (MM) cells are transformed from plasma B cells.

**Figure 2 ijms-25-04644-f002:**
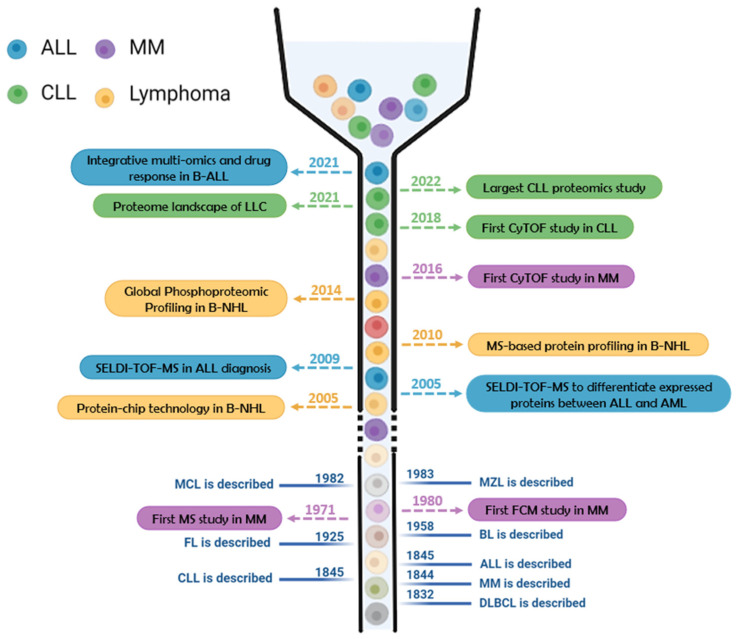
A vertical timeline of the most relevant hallmarks for B cell acute lymphoblastic leukemia (ALL, in blue), chronic lymphocytic leukemia (CLL, in green), multiple myeloma (MM, in purple), and lymphomas (in yellow), including the year when each malignancy was described (in blue) and meaningful protein-related discoveries. BL, Burkitt lymphoma; B-NHL, B cell non-Hodgkin’s lymphomas; CyTOF, cytometry by time of flight; DLBCL, diffuse large B cell lymphoma; FCM, flow cytometry; FL, follicular lymphoma; MCL, mantle cell lymphoma; MS, mass spectrometry; MZL, marginal zone lymphoma.

**Figure 3 ijms-25-04644-f003:**
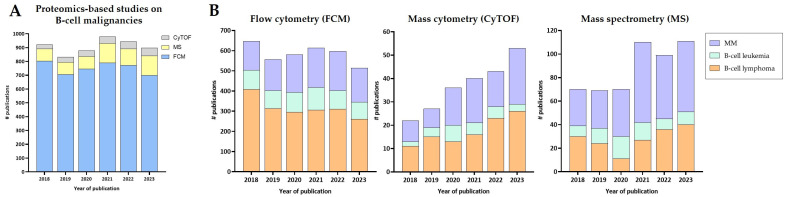
B cell-malignancy-related manuscripts published per year in the period from 2018 to 2023, employing flow cytometry, mass cytometry, and mass spectrometry as the main screening technologies. Section (**A**) shows the overall publication distribution per proteomics technique and year, whereas panels in (**B**) depict data specifying the publication rates for each B cell malignancy. MM, multiple myeloma.

**Table 2 ijms-25-04644-t002:** Summary of proteomic studies in B cell hematological malignancies.

Technology	Disease	Application	Reference
**Mass Spectrometry**	CLL	Correlation of genomic alterations with protein level variations	[77]
		Identification of a novel CLL subgroup associated with unfavorable clinical outcome	[78]
	B-ALL	Discovery of candidate biomarkers related to diagnosis, prognosis, and targeted therapy	[79]
		Definition of the role of IKZF1 alterations in ALL pathogenesis	[80]
		Development of a panel of candidate biomarkers for early diagnosis and treatment evaluation	[81]
		Identification of potential predictive markers of dexamethasone resistance in childhood ALL	[82]
		Study of the mechanisms involved in tolerance to vincristine	[83]
		Connection of the molecular phenotypes with drug responses and identification of therapeutic candidates for high-risk subtypes	[84]
		Screening of serum autoantibodies for early detection of B-ALL in children	[85]
	DLBCL	Characterization of pathways enriched in the different DLBCL subtypes	[86]
		Identification of proteins that discriminate between GCB and non-GCB lymphomas	[87]
		Finding of proteins differentially expressed between GCB and non-GCB subtypes in extracellular vesicles	[88]
	FL	Identification of differentially expressed proteins after p38 MAPK inhibitor treatment	[89]
		Distinguishing of proteins that interact with BCL6 and modulate its activity in transcriptional regulation	[90]
		Understanding nucleoside analogue resistance by quantification of intracellular araCTP, CTP, and dCTP	[91]
		Characterization of rituximab action mechanism	[92]
		Finding biomarkers for early disease detection and management	[93]
		Discovery of predictive indicators of histological transformation	[94]
	FL and MCL	Finding differentially expressed proteins between FL and MCL	[95]
	MCL	Identification of molecular signatures that differentiate MCL from B cells of the different compartments	[96]
		Identification of proteomic biomarkers to distinguish MCL	[97]
		Searching for specific proteomic biomarkers overexpressed in MCL tumor biopsies	[98]
		Identification of tyrosine-phosphorylated proteins	[99]
		Provision of insights into the dynamics of the disease and response to treatment	[99]
		Evaluation of resistance to antinucleoside drugs	[100]
		Discovery of neo-antigen peptides that mediate the killing of autologous lymphoma cells by circulating CD4 T cells	[101]
		Characterization of the action mechanism of the MDM2-antagonist nutlin-3a	[102]
	MZL	Discovery of the mechanisms involved in the pathogenesis of ocular adnexa extranodal MZL	[103]
		Identification of biomarkers for the diagnosis of primary Sjögren’s syndrome/MALT and prediction of progression	[104]
		Establishment of the role of ID3 in regulating cell proliferation	[105]
		Study of the pharmacokinetics of umbralisib	[106]
	BL	Analysis of differentially expressed proteins between endemic and sporadic BL variants and EBV^+^ and EBV^−^ BL cell lines	[107]
	MM	Prediction of MGUS progression for an early diagnosis of MM	[108]
		Analysis of the tumor microenvironment to identify determinants of durable responses to BCMA CAR T therapy	[109]
		Quantification of surface proteins to identify immunotherapy targets and biomarkers associated with resistance and response to treatment	[109]
		Identification of cell surface targets for immune-based therapies	[110]
**Flow Cytometry**	CLL	Design of panels for rapid disease diagnosis and progression assessment	[111,112]
		Comparison of residual normal B cell profiles between CLL and MBL	[113]
	B-ALL	Evaluation of neuropilin-1/CD304 as minimal residual disease and prognostic marker	[114]
	DLBCL	Assessment of the absolute counts of B cells, T cells, and Treg cells for the prognostication of newly diagnosed DLBCL patients	[115]
		Evaluation of the monocytic population distribution as an independent prognostic factor	[116]
	DLBCL & FL	Usage of aneuploidy and cell cycle indexing as tools for differentiating between CD10^+^ DLBCL and FL	[117]
	DLBCL & BL	Identification of cell markers to differentiate between BL and CD10^+^ DLBCL	[118]
	MZL	Distinguishing IgG4-related ophthalmic disease, idiopathic orbital inflammation, and extranodal MZL based on the expression of different markers	[119]
	MM	Prediction of response to daratumumab monotherapy based on baseline CD38 expression levels and CD38 reduction	[120,121,122]
		Identification of markers for optimal monitoring of granulocytic myeloid-derived suppressor cells	[123]
		Study of the PD-L1/PD-1 immune profile in patients with smoldering and active MM	[124]
		Identification of targets for CAR T cell therapy	[125]
		Characterization of global proteomes of CD3^+^, CD4^+^, and CD8^+^ T cells and development of a strategy to classify T cell subtypes	[126]
**Mass Cytometry**	CLL	Analysis of the tumor microenvironment to find differences in protein expression between tumor and normal cells	[127]
		Assessment of the healthy B cell pool of patients to find disease mechanisms	[128]
	DLBCL	Evaluation of the intertumoral and intratumoral heterogeneity	[129]
		Identification of clinically relevant interactions between tumor-associated macrophages and blood endothelial cells	[130]
		Finding proteins overexpressed in relapsed and refractory patients	[131]
		Comparison of major immune subsets in DLBCL and double-hit lymphoma	[132]
	MM	Identification of differences in immune cell compartments across various stages of MM and healthy individuals	[133]
		Description of the immune tumor microenvironment in patients with MGUS and MM at diagnosis and post-initial therapies	[134]
		Analysis of the immune checkpoint signature and regulation	[135]
		Characterization of NK cells in newly diagnosed cases	[136]
		Understanding of the molecular and cellular complexities underlying disease heterogeneity and prognosis	[137]
		Provision of insights into the mechanism of action of daratumumab and the anti-PD-L1 monoclonal antibody atezolizumab	[138]
		Employment of protein profiling as a tool for prognosis and treatment stratification	[139]
**Other Tools**			
**RRPA**	CLL	Prediction of survival outcomes based on the proteomic signature	[140]
**Protein Microarrays**	FL	Identification of antibodies that distinguish lymphoid follicles in FL and benign follicular hyperplasia	[141]
	MCL	Monitoring of patient serum proteomes to identify treatment-modulated proteins linked to the presence of minimal residual disease	[142]
**Western Blot**	MCL	Definition of the pathologic hallmark of MCL as a tool for the diagnosis	[143]

*araCTP*, 1-beta-D-arabinofuranosylcytosine triphosphate; *B-ALL*, B cell acute lymphoblastic leukemia; *BCL6*, B cell lymphoma 6; *BCMA*, B cell maturation antigen; *BL*, Burkitt lymphoma; *CAR*, chimeric antigen receptor; *CLL*, chronic lymphocytic leukemia; *CTP*, cytidine triphosphate; *dCTP*, deoxycytidine triphosphate; *DLBCL*, diffuse large B cell lymphoma; *EBV*, Epstein–Barr virus; *FL*, follicular lymphoma; *GCB*, germinal-center B cell-like; *ID3*, inhibitor of DNA binding 3; Ig, immunoglobulin; *MALT*, extranodal marginal zone lymphoma of the mucosa-associated lymphoid tissue; *MAPK*, mitogen-activated protein kinases; *MBL*, monoclonal B cell lymphocytosis; *MDM2*, murine double minute 2; *MCL*, mantle cell lymphoma; *MGUS*, monoclonal gammopathy of undetermined significance; *MM*, multiple myeloma; *MZL*, marginal zone lymphoma; *NK*, natural killer; *non-GCB*, activated B cell-like; *PD-1*, programmed death-1; *PD-L1*, programmed death ligand-1; *RRPA*, reverse phase protein array.

## Data Availability

Not applicable.

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
