# Peer review of "Characterization of Human B Cell Hematological Malignancies Using Protein-Based Approaches"

_ijms, 2024, doi:10.3390/ijms25094644_

Round 1

Reviewer 1 Report

Comments and Suggestions for Authors

The review discusses the role of protein-based technologies in characterizing human B-cell hematological malignancies, such as leukemias, myelomas, and lymphomas. It highlights the significance of proteomics strategies, including mass spectrometry, flow cytometry, and mass cytometry, in profiling molecular effectors, identifying biomarkers, and understanding disease mechanisms. The review also provides an overview of various B-cell malignancies, their clinical features, and treatment options, emphasizing the importance of these technologies in advancing research and improving patient care.

Strengths

Comprehensive Overview: The paper provides a detailed and comprehensive overview of B-cell hematological malignancies, covering their classification, molecular characteristics, and treatment options, which offers a thorough understanding of these complex diseases.

Focus on Proteomics: The paper highlights the importance of proteomics techniques, such as mass spectrometry, flow cytometry, and mass cytometry, in characterizing these malignancies at the molecular level, identifying biomarkers, and understanding disease mechanisms, which is crucial for advancing research and improving patient outcomes.

Summarization of Key Studies: The paper summarizes key proteomics studies conducted on different B-cell malignancies, providing insights into how these studies have contributed to our understanding of the diseases and the development of targeted therapies, showcasing the paper's role in synthesizing and presenting valuable research findings.

Areas to strengthen

Focus on intersection with other modalities: NGS Sequencing (RNA/DNA) is increasingly used in clinical practice. It would be useful to highlight the areas where Proteomics is differentiated and remains the best tool to use. Additionally, how can proteomics be synergistically be applied together with seqeuncing approaches to better classify and treat patients.

Limited Discussion on Future Directions: A more detailed discussion on the future directions of proteomics research in B-cell malignancies, including emerging technologies, challenges in translating research into clinical applications, and potential strategies for overcoming these challenges.

Inclusion of Case Studies or Patient Outcomes: The review could be enhanced by incorporating case studies or real-world examples of how proteomics has impacted patient outcomes in B-cell malignancies, providing a more practical perspective on the significance of these technologies in clinical settings.

Comments on the Quality of English Language

easy to follow

Reviewer 2 Report

Comments and Suggestions for Authors

The manuscript is a pertinent review about studies that used flow cytometry, mass cytometry and mass spectrometry to study most of the B-cell hematological malignancies. It describes representative studies that show the scope of these technologies as tools to address cancer biology.

Mayor points:

1.       It is necessary to include information on B-cell precursor acute lymphoblastic leukemia (B-ALL) in adult and pediatric populations.  I think B-ALL is also a B-cell hematological malignancy. Why was not included in the review?

2.       The information in section 1 was heterogeneous for the different B-cell hematological disorders. For example, in some of them, the incidence was mentioned but in others this information was missing. Treatment options should be described in the same depth for all the disorders (in section 1.3, treatment was described in more detail than in sections 1.1 or 1.2.x). Also, as the review included flow cytometry, it is crucial to include a list of antigens and CD markers (CD19 or CD3, etc.) for each disorder (probably as a table), this information was included for some disorders but for others it is missing.

3.       In section 2 it is necessary to clearly describe and compare the rate of false identifications (false positives) for flow cytometry, mass cytometry and mass spectrometry.  

Round 2

Reviewer 2 Report

Comments and Suggestions for Authors

The manuscript has been improved. In my opinion, it can be accepted for publication